# Pseudorabies Virus Associations in Wild Animals: Review of Potential Reservoirs for Cross-Host Transmission

**DOI:** 10.3390/v14102254

**Published:** 2022-10-14

**Authors:** Aijing Liu, Tong Xue, Xiang Zhao, Jie Zou, Hongli Pu, Xiaoliang Hu, Zhige Tian

**Affiliations:** 1Yibin Key Laboratory of Zoological Diversity and Ecological Conservation, Faculty of Agriculture, Forestry, and Food Engineering, Yibin University, Yibin 644000, China; 2State Key Laboratory of Veterinary Biotechnology, Harbin Veterinary Research Institute, Chinese Academy of Agricultural Sciences, Harbin 150001, China; 3School of Mathematical Science, Harbin Normal University, Harbin 150001, China

**Keywords:** pseudorabies virus, wildlife, cross-host transmission, vaccination

## Abstract

Pseudorabies virus (PRV) has received widespread attention for its potential health effects on humans, wildlife, domestic animals, and livestock. In this review, we focus on PRV dynamics in wildlife, given the importance of wild-origin PRV transmission to domestic and farm animals. Wild boars, pigs, and raccoons can serve as reservoirs of PRV, with viral transmission to domestic livestock occurring via several routes, such as wild herd exposure, contaminated meat consumption, and insect vector transmission. Many endangered feline and canine species can be infected with PRV, with acute disease and death within 48 h. The first confirmed human case of PRV infection in mainland China was reported in 2017. Thus, PRV exhibits potentially dangerous cross-host transmission, which is likely associated with inappropriate vaccination, poor awareness, and insufficient biosecurity. Currently, no vaccine provides full protection against PRV in all animals. Here, we summarize the epidemiology and pathogenesis of PRV infection in wild, domestic, and farmed animals, which may facilitate the design of novel therapeutics and strategies for controlling PRV infection and improving wildlife protection in China.

## 1. Introduction

Belonging to the family *Herpesviridae* and subfamily *Alphaherpesvirinae* [1], the pseudorabies virus (PRV) is the causative agent of pseudorabies (PR) or Aujeszky’s disease, which can lead to acute infection and significant economic losses in pigs and other animals. The alphaherpesvirus genomes are divided into six classes (A–F) [2] based on the arrangement of repeat sequences (inverted and tandem repeats) and unique regions (short and long). The PRV genome belongs to the D class [3] and is approximately 140 kb in length.

Wild boars and pigs are the primary natural hosts of PRV and the only latent carriers [4]. PRV-related morbidity and mortality depend on various factors, such as animal health, viral strain, and infectious dose [5]. To date, PRV has been detected in various wild animals, including boars [6], rats [7], bears [8,9], raccoons [10,11], panthers [12], Iberian lynx [13], wolves [14], bats [15], and cats [16], as well as in domestic and farm animals, such as dogs [17,18,19], cattle [20,21,22], cats [23], sheep [24,25], foxes [26,27,28], and minks [29,30,31,32]. Furthermore, PRV infection is also a potential threat to humans [33,34,35,36].

In this review, we provide a brief overview of the traceability, transmission, and vaccination of PRV infection in animals with the aim to strengthen PR control measures.

## 2. Possible Transmission Route of PRV Infection

At present, the epizootiology of PR disease is not completely understood. One hypothesis for herd-to-herd transmission is that wild animals (e.g., wild boars and Norway rats) act as disease reservoir hosts and spread the virus from one farm to another [37]. Alternatively, wild animals may become directly infected with PRV due to consumption of wild boar meat [38] or via insect-borne vectors such as winter ticks [39]. However, previous research on infected wild panthers found that the isolated PRV strain did not match that found in wild boars [12]. Thus, the origin of PRV in wild animals remains unresolved. In this review, we summarize current knowledge of PRV infection in natural and non-natural hosts (Figure 1).

## 3. PRV Infection in Wild, Domestic, and Farm Animals

### 3.1. PRV Infection in Wild Animals

#### 3.1.1. Wild Boars

Wild boars are reservoirs for various infectious diseases in livestock and humans, including classical (CSFV) and African swine fever viruses (ASFV), porcine circovirus 2 (PCV2), porcine reproductive and respiratory syndrome virus (PRRSV), porcine parvovirus (PPV), hepatitis E virus (HEV), swine influenza virus (SIV), Japanese encephalitis virus (JEV), and torque teno virus (TTV) [6,40].

PRV infection is enzootic and widespread in Eurasian wild boars [41,42], with the first cases reported in Italy, former Yugoslavia, and USA [43]. Following successful prevention and eradication programs, PRV has been largely eliminated from domestic pigs in many parts of Europe, including Austria, Cyprus, Denmark, Finland, The Netherlands, and Great Britain (England, Scotland, and Wales) [41]. However, PRV infections remain widespread in many wild boar populations in the Czech Republic [44], Germany [41,45], France [46], Croatia [47], USA [48,49,50], Japan [51], Italy [16,52,53], Switzerland [52], Spain [53,54], Slovenia [55], Serbia [56], Brazil [57], Romania [58], Russia [59], Poland [60], and Belgium [61]. To date, however, there is little epidemiological information regarding PRV infection in Chinese wild boar populations. Considering that the number of wild boars in China is increasing year by year, epidemiological surveillance of PRV is urgently needed.

As wild boar populations expand their range, there is increased potential risk for disease transmission that may affect healthy of humans and domestic swine, and conservation of wildlife. Based on long-term serological field studies, PRV is considered endemic in wild boar populations in the USA, including in the southern and central states (California [62], Southeast [63], Hawaii, Texas [64], Arizona [65], North Carolina [66,67]), as well as in Guam [68], Nebraska, Michigan [69], and Oklahoma [49]. Seroprevalence varies from 0.5% to 64.4% at the regional level [68,70,71]. In Europe, PRV-infected wild boar populations are present in Germany, Italy, Belgium, Croatia, the Czech Republic, former Yugoslavia, France, the Netherlands, Poland, Romania, Russia, Slovenia, Spain, and Switzerland, with average seroprevalence ranging from 0.3% to 66%. Based on 7209 samples collected from 2000 to 2011, average seroprevalence in Germany is 6.8% [45]. Based on 5000 wild boar samples collected from 2006 to 2020, average seroprevalence in Italy ranges from 3.8% to 30.9% [70,71,72,73].

Characterization of PRV from wild swine is helpful to understand population diversity and could trace back the infection route [43]. In Europe, phylogenetic analysis of partial sequences of the glycoprotein C gene suggests that wild boar isolates can be differentiated into clades A and B [61,74]. Clade A isolates originate from Austria, France, Germany, Hungary, Italy, and Slovakia, while clade B isolates originate from southwestern Europe, including Germany, France, and Spain [74]. Thus, the clade A and B isolates overlap geographically in central Europe, Germany, and France [74]. PRV isolated from USA was distinct from European isolates and was closely related to domestic pig isolates. It may represent a transmission from domestic to feral swine [6,75,76].

#### 3.1.2. Bears

Detection of antibodies to canine distemper virus (CDV), canine herpesvirus type 1 (CHV-1), canine parvovirus type 2 (CPV-2), canine parainfluenza virus type 2 (CPIV-2), canine coronavirus (CCV), West Nile virus (WNN), and PRV is positive in European brown bears (*Ursus arctos*) [77]. Previous studies have reported PRV-positive blood samples in free-ranging Eurasian brown bears (*Ursus arctos*) in Slovakia [77] and PRV-positive tissues and antibodies in captive bears fed raw pig heads [8,9,76,77].

In most cases, PR disease is fatal in bears, with death occurring within 1–3 days of clinical symptoms. Upon consuming infected pig offal or pig heads, bears can show acute and asymptomatic infections. In Himalayan bears, however, only mild clinical signs are reported, with negative neutralizing antibody tests [8].

#### 3.1.3. Raccoons

Raccoons (*Procyon lotor*) are considered a natural reservoir of PRV [10]. The clinical signs of PRV infection in raccoons resemble those of rabies and distemper [78]. Avirulent strain K is naturally occurring in the raccoon population [79]. Raccoons infected with virulent PRV strains (e.g., Be, S62/26, and 429) can transmit the virus to uninfected raccoons via contact. In raccoons, all virulent PRV strains are lethal, while avirulent strain K has a mortality rate of only 18% (2/11). Furthermore, when raccoons are infected with avirulent strain K, they may be re-infected with virulent strains, suggesting that virulent PRV may be circulating in raccoon populations [10,78].

#### 3.1.4. Other Wild Animals

PRV not only infects wild boars, bears, and raccoons, but also can infect many wild animals, especially rare and endangered animals [12,53].

##### Panthers

The Florida panther (*Felis concolor coryi*) is an endangered feline species [12]. Similar to the infection process in domestic cats, PRV infection is fatal in panthers [80]. Necropsies of infected panthers have detected residual swine hair in the digestive tract, confirming ingestion of wild boar meat and identifying the potential route of PRV exposure. Based on sequence analysis, the isolated panther strain has been identified as a wild-type gI^+^ and TK^+^ genotype [81].

##### Iberian Lynx

The Iberian lynx (*Lynx pardinus*), one of the most endangered feline species in the world [53], preys on small ungulates, birds, reptiles, and occasionally wild boar [82] and is positive for feline leukemia provirus (FeLV), feline parvovirus (FPV), and *Cytauxzoon* sp. by using PCR assay [13]. A previously captured lynx was found to be PRV-positive based on blood, oropharyngeal swab, and rectal swab samples, with PRV antigens also detected in the tonsils, brain, and gastric glandular epithelial cells [53]. Histopathological analysis of the central nervous system (CNS) further showed meningoencephalitis similar to that reported in domestic cats [83,84], dogs [85,86,87], foxes [88], and coyotes [89]. These results suggest that the Iberian lynx is susceptible to PRV and may be exposed to the virus via the consumption of wild boar meat [82]. Molecular and seroepidemiological analysis further confirmed the emergence of PRV in the Iberian lynx population (11.8%, 2/17) [13].

##### Coyotes

PRV infection has been reported in coyotes (*Canis latrans*) based on the testing of brain tissue. Clinical symptoms include anorexia, abnormal vocalizations, and nervousness [14,87,88]. PRV isolated from Belgian coyotes is similar to the Kaplan reference strain, the most common type present in European wild boars, suggesting potential infection via consumption of PRV-infected boar meat [14]. In China, a PRV strain was isolated from a coyote showing various clinical symptoms, including vomiting, dyspnea, circular movements, processed moaning, uneasy behavior, intense pruritus, paroxysmal convulsions, and quadriplegia. Sequence analysis of the Chinese PRV strain, which contained the glycoprotein E gene, indicated that it may be a field strain rather than a vaccine strain [90].

### 3.2. PRV Infections in Domestic Animals

#### 3.2.1. Dogs

Because of the recent incidents of PRV infecting people, the dog in PRV infection has aroused great concern. PRV is prevalent in dogs, including working and hunting dogs, in the USA [17,89,90], Belgium [91], Italy [19,75,92], France [93], Spain [85,94], Japan [95,96], Austria [97,98,99], China [18], Serbia [100], Argentina [101], and Germany [74], with infection via direct or indirect contact with wild boars and pigs [6,74]. Dogs with PRV infection display neurological signs, anorexia, intense muzzle itch, and respiratory distress [1,96], and usually die within 48 h of clinical symptom onset [19].

#### 3.2.2. Cattle

PRV infection can occur in cattle [6,20,21], especially when housed with pigs. PRV-infected cattle present with neurological symptoms and usually die within 24 h, with necropsy showing leptomeningeal hyperemia and lung consolidation [21]. Furthermore, under experimental infection, PRV has been isolated from the retropharyngeal lymph nodes [102] and pituitary, pharynx, and submaxillary lymph nodes of cattle [103].

#### 3.2.3. Cats

Cats can be infected with PRV under natural and experimental conditions [81,82,103]. The disease is sporadic and mainly transmits through the consumption of pig offal. Clinical symptoms include anorexia and pruritus, with death occurring within 12–48 h of clinical onset [83]. The virus is transmitted via the oral route, typically replicating in the tonsils and pharynx, then spreading through the CNS and excreted through oral and nasal secretions [83,104].

#### 3.2.4. Sheep

Sheep in contact with pigs are also susceptible to field strains of PRV. Previous studies have reported clinical signs of lethargy, fever, pruritus, and death in sheep following PRV infection, despite vaccination with the Bartha vaccine strain [24]. Even without direct contact with pigs, sheep exhibiting CNS symptoms have been found to be PRV positive in postmortem tissue samples, with sequence analysis of the viral genome showing close identity to the Buk T-900 reference strain [105]. China has also reported outbreaks of PR in sheep following vaccination with attenuated live vaccines (i.e., Bartha-K16), resulting in clinical symptoms such as intense rubbing and licking. These results suggest that certain vaccines may be unsuitable for use in sheep [25].

#### 3.2.5. Horses

Previous reported that horses with experimentally induced PRV infection had severe clinical signs, but no pruritus [106]. PRV antigens and DNA have also been detected in the neurons of horses presenting with neurological signs and severe meningoencephalitis, with PRV infection potentially occurring via aerosolized pig slurry [106]. While other non-porcine species typically succumb to PRV infection within 48 h, PRV-infected horses exhibit a longer course of infection (more than 7 days) without pruritus, and eventually survive [106,107].

#### 3.2.6. Chickens

In addition to infecting mammals, PRV can also infect birds such as chickens, which are susceptible to PRV infection under both experimental and natural conditions [108,109,110]. For example, Kouwenhoven (1982) isolated a PRV strain from farm chickens, which was pathogenic to 7-day-old chicks but could be neutralized by known PRV antisera and showed cross-reactivity with infectious bovine rhinotracheitis. The strain was originally derived from a PRV vaccine adapted to chicken cells and did not cause classical inclusion bodies in the neural tissue [110].

### 3.3. PRV Infections in Farm Animals

#### 3.3.1. Foxes

Foxes fed pig offal can become infected with PRV, as reported in Italy [26,28] and China [27]. Foxes with PRV infection display neurological signs, ataxia, fever, vomiting, dyspnea, intense pruritus, and frequent snarling, with death usually occurring within a few hours to three days. Studies have reported a high morbidity rate (80%; 1200/1500) in PRV-infected foxes, especially symptomatic foxes [27]. Previous epidemiological analysis of strains isolated from dead foxes found a close relationship to domestic field strains in China, rather than vaccine strains [27]. However, the epidemiological link between the isolated strain and wild animal populations or domestic pigs is unclear.

#### 3.3.2. Minks

Outbreaks of PRV in minks have been reported in many countries [111,112,113,114,115]. Brain samples collected from dead minks have been identified as PRV-positive, with necropsy also showing organ hemorrhage and endotheliotropic vessels, although no neurological signs were evident before death [29,30]. In 2014, a serious outbreak of PRV was reported in a mink farm in Shandong Province, China. The minks exhibited severe clinical signs, including diarrhea, anorexia, and abdominal and facial skin scratching, and a high morbidity rate of 87% (3522/4028). Out of 566 minks, 33 tested PRV-positive by polymerase chain reaction (PCR), and the isolated strain clustered with vaccine-resistant Chinese porcine PRV isolates [31,32].

### 3.4. PRV Infections in Humans

Although humans were previously considered a non-susceptible host for PRV infection, a PRV variant causing infectious endophthalmitis has been reported in China in recent years [36], thought to have originated from sewage containing pig excrement. In 2020, four patients with acute encephalitis were diagnosed with PRV infection and presented with respiratory dysfunction and acute neurological symptoms. The PRV strain (hSD-1/2019) isolated from the cerebrospinal fluid of one of the patients was closely related to a PRV variant known to cause high pathogenicity and acute neurological symptoms in pigs [35].

## 4. Transmission Models and Cross-Host Transmission Routes for PRV

There are two competing models of PRV transmission in wild boars, i.e., age-dependent and sexual transmission models. The age-dependent model suggests that PRV seroprevalence differs between older and younger animals, with older boars showing significantly higher prevalence than younger individuals [16,116,117,118,119]. In contrast, preferential sexual transmission may occur under random or polygynous boar mating systems [48], whereby polygamy and mate guarding behavior may impact the seroprevalence of PRV in males and females differently. For example, under a 1:1 sex ratio, predicted seroprevalence is the same for both at-risk males and at-risk females; however, as wild boars are highly polygynous [120,121], the predicted seroprevalence is different for at-risk males and at-risk females [48].

Although data on the origin and transmission of PRV field strains are limited, we suggest several possible transmission routes: (1) non-natural hosts may become infected via contact with immunized pigs [122]; (2) before succumbing to illness, PRV-infected non-natural hosts may travel within their territories, leading to transmission to other individuals; (3) due to inappropriate biosecurity measures, non-natural hosts may come into indirect contact with carcasses contaminated with PRV [9]; (4) as anthropogenic-impacted landscapes change wild animal habitat availability, frequent contact between wild animals may result in an increase in PRV infection. Thus, appropriate measures should be taken to disrupt these transmission routes and control animal infection.

## 5. Vaccination

PRV has a wide host range and can be transmitted across wild, domestic, and farm animals. The DIVA strategy (i.e., differentiating infected from vaccinated animals) has been used in various European countries and the USA based on gE- or gI-deleted vaccines [67,72]. Attenuated live vaccines (e.g., gE-gene, gE/gI/TK-gene, TK/gG-gene, gE/gI-gene, TK/gE-gene and gE/US2-gene deleted) have been used in China since the early 1990s for the control of PR in pig farms [123,124,125,126,127]. However, while these vaccines provide effective protection for pigs, they do not appear to provide full protection for all animals or species. Previous reports suggest that virulent strain DCD1 and attenuated strain HB98 can cause clinical signs and pathological lesions and induce death within 24 h of the onset of symptoms [128]. Interestingly, in transmission experiments using the Bartha-K61 vaccine, viruses with longer incubation periods and lower replication rates take longer to cause lesions in dogs than the HB98 and DCD-1 strains [128]. In addition, US7/US8/UL23-deleted recombinant PRV vaccines provide effective protection in pigs but show obvious neural symptoms and virulence in dogs [129], while the Bartha-K16 vaccine can also induce PR in sheep and goat populations [25]. Thus, commercial vaccines may not be safe for dogs or certain farm animals [129]. These findings suggest that genes deleted in commercial vaccines may be associated with replication rates and tissue tropism in vivo, resulting in changes in PRV pathogenicity [128]. In contrast, modified live PRV vaccines with gG-gE/TK deletion have been shown to induce an immune response against the virulent Shope strain and may play a role in preventing PRV transmission in raccoons [11]. In general, however, present-day vaccines used in pig farms do not provide complete protection for all farm and domestic animals. As such, novel and safe vaccines must be developed for different animals, particularly those that are endangered.

## 6. Interspecies Transmission

PRV was previously considered to be limited to host species only. However, there are three possible pathways that have allowed cross-host transmission from wild boars and pigs to non-natural hosts, namely, positive selection, adaptive evolution, and recombination. The *gB*, *gC*, *gD*, and *gE* genes play central roles in receptor binding, pathogenicity, and induction of antibodies, with various residues shown to be under positive selection, including residues 43, 75, 848, and 922 in the gB protein, residues 59 and 194 in the gC protein, and residue 348 in gE protein [130], thus suggesting that PRV may undergo host adaptation. Further research has revealed that two sites on gB in clade 2 of PRV (929 and 934), four sites on gC in clade 1 of PRV (59, 75, 76, and 191), and two sites on gE in clade 2 of PRV (495 and 540) may be related to adaptive evolution after cross-host transmission [130]. Thus, these two pathways suggest that mutations on the PRV glycoprotein may play a role in facilitating cross-host transmission from natural to non-natural hosts (including humans). Recombination also plays a vital role in the evolution of viruses, allowing the emergence of new strains with altered virulence and immunogenicity [131]. Intraclade and interclade recombinations have occurred in the PRV genome. For example, the SC strain is thought to have recombined with PRV strains in clade 2 and the PRV ZJ01 strain is thought to have originated by recombination between clade 1 or clade 2.1 isolates [132]. These recombinations can cause enhanced virulence, failed immunity, or new genotypes [130], and may increase the probability of cross-host transmission of PRV.

In addition, the receptors of PRV have been identified, including 3-O-sulfonated-heparan sulfate (3-O-S-HS) [133], the herpes virus entry mediator A (HveA, also known as HVEM), a TNF receptor-related protein [134], and three immunoglobulin superfamily members: HveB (PRR2, nectin-2) [135], HveC (PRR1, nectin-1) and HveD (PRV, CD155) [136]. Among these molecules, nectin-1 is highly conserved in mammalian animals and serves as a broadly used receptor mediating the entry of PRV [136]. Previous reports suggested that PRV infects host cells via both human and swine nectin-1, and that its gD exhibits similar binding affinities for nectin-1 of the two species [137]. These structural observations provide a systematic view on the receptor binding mechanism for cross-host transmission of PRV.

## 7. Conclusions

In this review, we summarize the epidemiology and pathogenesis of PRV infection in wild, domestic, and farm animals. Recently, various field strains have shown virulence against non-natural hosts, such as bears, coyotes, and panthers, with some found to be epidemiologically associated with swine isolates, wild boar isolates, and vaccine strains. Thus, novel and safe vaccines should be developed for different animals, particularly endangered species, to control cross-host transmission of PRV.

## Figures and Tables

**Figure 1 viruses-14-02254-f001:**
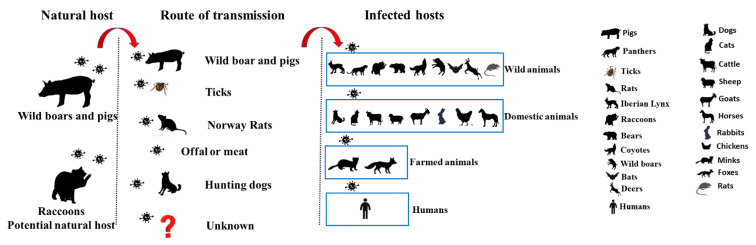
Schematic of PRV transmission. Wild boars and pigs are natural hosts of the PRV, while raccoons are potential natural hosts of the PRV. Five known transmission routes are described. Infected hosts include humans and wild, domestic, and farm animals.

## Data Availability

All data generated or analyzed during this study are included in the published article.

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
