# Peer review of "Pseudorabies Virus Associations in Wild Animals: Review of Potential Reservoirs for Cross-Host Transmission"

_viruses, 2022, doi:10.3390/v14102254_

Round 1

Reviewer 1 Report (Previous Reviewer 3)

The author have revised the manuscript according to the comments, the manuscript has been well organized and can be published.

Reviewer 2 Report (Previous Reviewer 2)

Dear author, more details of the cross-host transmission have added in this revision, which have fully addressed our concerns. And we believe that this manuscript will attract many other researchers interest.

Reviewer 3 Report (Previous Reviewer 1)

The title of the manuscript is very attractive, and the manuscript has good academic value.

This manuscript is a resubmission of an earlier submission. The following is a list of the peer review reports and author responses from that submission.

Round 1

Reviewer 1 Report

The title of the manuscript is very attractive, and  the manuscript has good academic value.

Reviewer 2 Report

Dear authors, I still recommend rejecting this manuscript. Because that reply did not adequately address my concerns and did not elaborate on the topic of cross-host transfers. Thank you for your attention to my review, but unfortunately this manuscript is still below expectations for a peer-reviewed publication. I also believe your comments will be of interest to other journals. I wish you all the best and I do hope to see your comment finally published.

Reviewer 3 Report

I have checked the resubmitted munuscript, the present form was improved, and I agree to accept the munuscript.